# The Potential Role and Regulatory Mechanisms of MUC5AC in Chronic Obstructive Pulmonary Disease

**DOI:** 10.3390/molecules25194437

**Published:** 2020-09-27

**Authors:** Jingyuan Li, Zuguang Ye

**Affiliations:** 1Tianjin State Key Laboratory of Modern Chinese Medicine, Tianjin University of Traditional Chinese Medicine, Tianjin 301617, China; lijingyuan886@126.com; 2Institute of Chinese Materia Medica, China Academy of Chinese Medical Sciences, Beijing 100700, China

**Keywords:** COPD, muco-obstructive lung diseases, airway mucus hypersecretion, MUC5AC, cell differentiation

## Abstract

Chronic obstructive pulmonary disease (COPD) is associated with high morbidity and mortality globally. Studies show that airway mucus hypersecretion strongly compromises lung function, leading to frequent hospitalization and mortality, highlighting an urgent need for effective COPD treatments. MUC5AC is known to contribute to severe muco-obstructive lung diseases, worsening COPD pathogenesis. Various pathways are implicated in the aberrant MUC5AC production and secretion MUC5AC. These include signaling pathways associated with mucus-secreting cell differentiation [nuclear factor-κB (NF-κB)and IL-13-STAT6- SAM pointed domain containing E26 transformation-specific transcription factor (SPDEF), as well as epithelial sodium channel (ENaC) and cystic fibrosis transmembrane conductance regulator (CFTR)], and signaling pathways related to mucus transport and excretion-ciliary beat frequency (CBF). Various inhibitors of mucus hypersecretion are in clinical use but have had limited benefits against COPD. Thus, novel therapies targeting airway mucus hypersecretion should be developed for effective management of muco-obstructive lung disease. Here, we systematically review the mechanisms and pathogenesis of airway mucus hypersecretion, with emphasis on multi-target and multi-link intervention strategies for the elucidation of novel inhibitors of airway mucus hypersecretion.

## 1. Introduction

COPD is characterized by high mortality and morbidity globally. The World Health Organization (WHO) predicts that COPD will become the third among the top causes of deaths by 2030 [1]. In 2017, 3.2 million deaths were attributed to COPD, globally. A 2012–2015 Chinese epidemiological survey found the prevalence of COPD in individuals older than 40 years to be 13.7% [2]. COPD progression leads to a significant reduction in the quality of life, worsening the healthcare burden.

COPD is considered a treatable and preventable disease which often presents with unyielding airflow limitation. It is often driven by chronic inflammatory response of the airways and lung tissue. The complex pathogenesis of COPD involves protease-antiprotease imbalance, oxidative stress and chronic inflammation. The disease is characterized by irreversible airway remodeling that involves thickening of the airway wall and the airway smooth muscle (ASM) layer, mucus hypersecretion and epithelial cells metaplasia [3]. Thus, COPD is also regarded as a muco-obstructive disease accompanied by increased amounts of mucin, particularly MUC5AC that may uniquely modify mucus properties [4].

COPD risk factors include air pollution (PM2.5), tobacco smoking, exposure to biomass fuel, as well as genetic susceptibility. The 2018 China pulmonary health [CPH] epidemiological survey found that a smoking rate exceeding 20 pack-years increases the prevalence of COPD by 2-fold. Air pollution affects the health of populations in developing countries, such as China. Long-term and short-term exposure to PM2.5 contributes to the development of COPD. Heavy exposure to PM2.5 increases the occurrence of COPD by 2-fold in never-smokers and in the general population in China [2]. Consumption of biomass fuels increases indoor air pollution which contributes to the development of COPD, according to other studies. Additionally, epigenetic and genetic factors have been implicated in the pathogenesis COPD.

Efforts to control the public health effects of COPD include developing preventive or mitigation measures to improve patients’ quality of life [5]. Treatment of patients with stable COPD include long-term oxygen therapy, pulmonary rehabilitation, and smoking cessation. However, pharmacological agents are still the main for treatments for all COPD patients after smoking cessation, which are mainly inhaled. The long-acting muscarinic antagonist (LAMA) is the recommended initial treatment choice for patients with mild disease and no exacerbations [6]. In case of more severe dyspnea, severe airflow obstruction, and lung hyperinflation, combining LAMA with a selective long-acting beta2-agonist (LABA) is recommended [7,8,9,10]. If the symptoms and exacerbations persist (more than two exacerbations per year or one hospitalization for COPD), a LAMA, a LABA, and an inhaled corticosteroid (ICS) may be prescribed [11,12,13,14]. An array of other systemic therapies (azithromycin, roflumilast, xanthines, and antioxidants) may be considered as third-line treatments [1]. The use of antibiotics in for COPD exacerbations remains controversial [15].

Although smoking cessation, lung volume reduction surgery, oxygen inhalation, and drug therapy improve pulmonary function to an extent, reduce the frequency of acute exacerbation of chronic obstructive pulmonary disease (AECOPD) and enhance quality of life, they do not halt COPD progression [16]. Multiple studies indicate that airway mucus hypersecretion contributes to the rapid deterioration of lung function and progression of COPD, including hospitalization and mortality. COPD with airway mucus hypersecretion increases the risk of deaths by 3.5-fold relative to non-airway mucus hypersecretion disease [17].

Mucins have been linked to mucus hypersecretion [18,19]. Among them, MUC5AC is considered the most important one as it is overproduction in COPD results in mucus obstruction. Thus, MUC5AC is essential to the pathogenesis of COPD [20,21,22]. In this review, we discuss MUC5AC function, regulatory mechanisms, pathophysiological effects, and clinical significance in COPD.

## 2. The Biological Characteristics of Mucins

### 2.1. Classification of Mucins

The main airway mucus secretory cells are airway goblet cells, submucosal glands, and Clara cells. Ciliated cells are responsible for mucociliary clearance. All airway secretory cell types differentiate from airway epithelial cells [23,24,25]. The airway mucus consists of a sol layer and a gel layer that together form a defense barrier termed the “cilia-mucus blanket” [26,27]. The sol layer is mainly comprised of water and is close to the surface of epithelial cells. The gel layer, which lies above the sol layer, consists of water, mucins, lysozyme, lactoferrin, and various peptides, including mucin; the main source of mucoelasticity. Mucin may be membrane-bound and secreted [28]. Mucins fall into four classes based on structure (Table 1):(1) membrane-constrained mucins which contain tandem repeats (TR) include Muc20, Muc17, Muc18, Muc16, Muc15, Muc13, Muc12, Muc3B, Muc1, Muc3 and Muc4 [29,30,31]. (2) Cysteine-rich secretory mucins (containing TR sequences and include Muc6, Muc19, Muc5B, Muc5AC, and Muc2 [32,33]. (3) Cysteine deficient secretory mucins (containing TR sequences) consist of Muc7, Muc8 and Muc9 [30]. (4) Mucins without TR sequence include Muc18, Muc15, and Muc14 [33,34,35,36].

Cysteine-rich secretory, or gelatinous secretory mucin gives mucus its characteristic viscoelasticity and this property is critical to clearance of xenobiotics by ciliated cells. MUC5AC and MUC5B are the most important and abundant mucins, comprising about 75% of mucins [4,37]. These mucins are abundantly expressed in normal and stressed states. MUC5AC is the 1st line macromolecular substance, and represents 95% of total mucin secretion in the airway epithelium. However, abnormal quantity and quality of MUC5AC negatively affects airway function and may cause serious airway disease. MUC5AC is mainly secreted by the goblet airway epithelium, and its expression changes significantly under stress such as air pollution (PM2.5), tobacco smoking, as well as exposure to biomass fuel [30,38].

### 2.2. Mucin Secretion and Transport

#### 2.2.1. Airway Epithelial Cell Differentiation

The airway epithelium is comprised of the cartilaginous trachea that branches into primary bronchi, bronchioles, and alveoli, which gaseous exchange sites [39]. The tracheobronchial epithelium consists of multiple cell types, such as secretory, basal, and ciliated cells covering the basement membrane [40] (Figure 1). Under normal physiological conditions, the pseudostratified ciliated bronchial epithelium is comprised of normal proportions of goblet, ciliated and basal cells [40,41]. External insult, for example by tobacco smoke or microbial invasion, triggers a hyperplastic response by goblet cells, a decline in the number of ciliated cells, and ciliated cell dysfunction [42]. This transformation results in over-production of mucus and plays an important role in the COPD development.

#### 2.2.2. Mucus Hydration

To maintain viscoelasticity, mucus secreted into the extracellular space through exocytosis must absorb >100 times its mass in water. Because water passively shuttles through the cell membrane along the ion current, its amounts are determined by Cl^−^ release and Na^+^ uptake in the airway. In human airways, ENaC-mediated Na^+^ absorption is a critical regulator of mucosal hydration, leading to effective mucus clearance [43] (Figure 2). In healthy individuals, well balanced epithelial Na^+^ absorption and Cl^−^ secretion hydrates airway surfaces, promoting efficient mucociliary clearance (MCC) [44]. In the presence of muco-obstructive lung disease, an ion transport imbalance coupled with mucin hypersecretion elevates mucin concentration in the mucus layer, osmotically compresses the periciliary layer (PCL), and compromises MCC [45,46]. Moreover, other ion channels, including CFTR, Ca^2+^-activated Cl^−^ channel (CaCC) and Slc26A9, affect Cl^−^ release [47]. Additionally, oxidative stress is known to have acute and chronic effects on airway ion transport [48].

#### 2.2.3. Mucus Transport

Ciliated cells are responsible for the proper mucus transport. This process depends on the number of ciliated cells and adequate ciliated cells function, including CBF, ciliary waveform, and cilia orientation [49]. Ciliated cells, which have about 200 cilia each, swing cilia in forward in a directionally coordinated manner, driving the removal of microorganisms and particulate matter trapped by the mucus layer from airways. Thus, ciliated cells are constantly beating, pushing mucus up and out through throat (center). Hence, cell differentiation, hydration and ciliary activity are the main determinants of mucus clearance [50] (Figure 2).

## 3. Clinical Significance of Mucin in COPD

### 3.1. Mucus Constitutes a Natural Physical Barrier

Airway epithelial cells (AEC) protect the lung by acting both as a mechanical and an immunological barrier. The epithelial mucus layer is the 1st line of defense against bacterial, viral and other toxicants (tobacco smoke, biomass fuel, air pollution) invasion [51,52]. Through its dense network of tight junctions and transcellular adherens, AEC function as a physical barrier between the body and the environment. Hydration and cross-linking of mucins and mucin concentration determine mucus viscoelasticity and thus the firmness of the mucus mechanical barrier. Alterations in mucus surface hydration and mucus concentration occur with inhalation of large particles, as well as bacterial and viral infections. Mucus traps particles and a ciliary escalator that transports it towards the oropharynx for elimination through expectoration or swallowing. This system constitutes the 1st line of innate defense, and is considered the host defense mechanism for the lungs. This process is called mucociliary clearance [53,54].

### 3.2. Mucus Hypersecretion Increases Airway Resistance

Since airway area is the main determinant of airway resistance, the larger conducting airways are the chief site of airway resistance. Mucus hypersecretion increases the size of the mucus plug and causes its retention in the airways, reducing airway area. Along with the thickened airway walls, the mucus plugs increase airway resistance and decrease airflow through the lungs [55,56]. COPD patients manifest with elevated mucus secretory cells in the large airways and mucus accumulation in the epithelium and lumen of small airways. The level of mucus accumulation in the lumen of small airways closely correlates with the severity of airflow limitation in most COPD patients. Retention of small airway mucus and even formation of mucus plugs have been observed in the lung tissue of patients who have succumbed to COPD [57,58]. Patients with small airway mucus obstruction do not exhibit symptoms like chronic cough and expectoration, probably because they lack cough receptors [47]. Thus, some COPD patients exhibit airway mucus hypersecretion, aggravated airflow obstruction, high airway resistance, chronic cough, and expectoration.

### 3.3. Mucus Hypersecretion Contributes to Pathogen Colonization

Mucins are highly glycosylated, which not only contributes to the viscoelasticity of mucins but is also though to mediate host-pathogen interactions. Alterations in mucin glycan pattern may result in bacterial colonization [59,60]. MUC5AC is a huge multidomain oligomeric secretory molecule. It consists of a heavily O-glycosylated apoprotein core and extensive intramolecular disulfide bonds between the cysteine-rich amino and carboxy-sequence, and 7 potential N-glycosylation sites. This structure facilitates easy MUC5AC coupling with unique sialylation and sulfation patterns that contribute to host immunity and bacterial colonization. Highly sulfated mucus causes acidification of the airway microenvironment, which significantly reduces the bactericidal efficacy of commonly used antibiotics [61]. This makes it difficult to clear pathogenic bacteria following lower respiratory tract infection, resulting in colonization by the pathogen, persistent infection, and further mucus hypersecretion. These events promote airway mucus retention and viscosity, making hypersecretion an independent risk factor for COPD progression and mortality [62].

## 4. Signaling Pathways Associated with Airway Mucus Hypersecretion

Various signaling pathways have been implicated in the aberrant secretion and production of MUC5AC and can be grouped into the following based on mucin secretion and transport process: (1) signaling pathways associated with mucus secretion of cell differentiation. These include NF-κB and IL-13-STAT6-SPDEF signaling [63,64,65]. (2) Ion channel signaling pathways associated with mucus components. These include ENaC, CFTR, and oxidative stress pathways [66,67,68]. (3) Signaling pathways associated with mucus transport and excretion-CBF [69].

### 4.1. Signaling Pathways Associated with Mucus Secretion of Cell Differentiation

#### 4.1.1. NF-κB Pathway

In COPD, extracellular signaling factors such as tumor necrosis factor-α (TNF-α) bind to cell surface receptors to induce a cascade of downstream responses. Receptor stimulation triggers IκB kinase (IKK) activation. IKK then phosphorylates serine at the regulatory site of the IκB subunit of the intracellular NF-κB, allowing the IκB subunit to be ubiquitinated and degraded [70,71,72]. This in turn liberates NF-κB dimers, followed by translocation of the free NF-κB into the nucleus, where it drives the expression of target genes, including MUC5AC.

A study on a mucus hypersecretory cell model by Su Ui Lee et al. found that verproside significantly antagonizes the TNF-α-induced MUC5AC expression by inhibiting NF-κB expression and phosphorylation of its upstream effectors, including IKKβ, TGF-β-activated kinase 1 (TAK1) and IκBα in NCI-H292 cells [73]. A study by Soyoung Kwa et al. revealed that resistin modulated the expression of mucin in human airway epithelial cells through mitogen-activated protein kinase (MAPK)/NF-κB signaling. However, NF-κB inhibition suppressed resistin-induced MUC5AC expression, but not MUC5B expression [74] (Figure 3).

#### 4.1.2. IL-13-STAT6-SPDEF Pathway

SPDEF is an important member of the E26 transformation-specific (ETS) family of transcription factors. SPDEF was initially thought to regulate prostate development and prostate cancer development [75]. However, recent studies show that SPDEF is also expressed in the respiratory system, where it modulates the differentiation of airway epithelial cells [76]. The IL-13-STAT6-SPDEF-MUC5AC signaling pathway is the main signaling pathway driving cell differentiation of epithelial cells into goblet cells. A study by Seibold MA et al. found that SPDEF knockdown decreased IL-13-induced MUC5AC expression in human airway epithelial cells and enhanced Forkhead box A2 (FOXA2) expression, a transcription factor known to prevent mucus production [77]. A high throughput screening for transcriptional networks modulating differentiation of cultured airway epithelial cells identified RCM-1, a small nontoxic molecule as an inhibitor of excessive mucus production and goblet cell metaplasia in mice after exposure to allergens. This molecule suppressed signaling by IL-13 and STAT6, downmodulating expression of the STAT6 target genes FOXA2 and SP*DEF,* which modulate goblet cell differentiation [78]. (Figure 3)

### 4.2. Ion Transport Pathways Associated with Mucus Hypersecretion

The main determinants of effective clearance of airway mucus are sufficient ciliary beat frequency and adequate mucus hydration. Airway surface hydration is mainly governed by ENaC and CFTR ion transport pathways [44,66]. Oxidative stress has been found to cause acute and chronic alterations to airway ion transport. Numerous studies have implicated that oxidative stress is a contributor to the damage of ion transport channels. Antioxidant drugs that reduce mucus viscosity by acting on the disulfide bonds of respiratory mucin oligomers facilitate ion transport [79].

Cigarette smoke contributes to COPD pathogenesis by modifying CFTR function. The CFTR protein functions as an anion channel on the apical surface of airway epithelia and mutation in CFTR cause aberrant Cl^-^ and Na^+^ transport, leading to airway mucus dehydration and hypersecretion. Impaired CFTR function compromises mucociliary clearance, leads to airway obstruction and chronic airway infection, with Gram-negative bacteria such as *P. aeuroginosa* causing respiratory failure [80]. CFTR belongs to the ATP-binding cassette (ABC) subfamily of transporters. ABC proteins bind ATP and use the energy to drive the transport of various molecules across cell membranes. In CFTR, the interactions of ATP and CFTR’s nucleotide-binding domains control the opening and closing of the channel, rather than driving solute transport. Deletion of F508 on CFTR causes its retention in the endoplasmic reticulum, reducing CFTR open probability, and its residence time in the plasma membrane [81,82]. Upon CFTR activation, Cl^−^ and HCO_3_^−^ are extruded into the airway lumen, with Na^+^ and H_2_O following passively through the paracellular pathway. (Figure 2)

Recent studies suggest that aberrant electrolyte transport as a result of impaired CFTR channel function fails to create the acidic environment needed for phagolysosome activity, providing conducive conditions for bacteria reproduction [83,84].

ENaC-mediated Na^+^ absorption across the apical plasma membrane modulates mucosal hydration, directly contributing effective mucus clearance. Cl^−^ secretion is facilitated by either CFTR, or CaCCs, including Anoctamin-1/TMEM16A. Ciliated cells co-express Anoctamin-1, CFTR, and ENaC, which clear mucus and maintain lung sterility [85]. ENaC is a heterotrimer of α, β, and γ-subunits. The α- and γ-subunits must undergo proteolytic cleavage to enable ENaC activation and Na^+^ conductance [86]. Ubiquitination of the α and γ ENaC subunits on their N-termini by the E3 ubiquitin ligase Nedd4-2 (NEDD4L) limits the channel’s half-life. Knockdown of NEDD4-2 in mice has been shown to upregulate ENaC, resulting in lung pathology. Upon ENaC activation, Na^+^ enters the blood from the airway lumen and Cl^−^/H_2_O moves in the same direction, reducing mucus volume [87]. Cl^−^ is the most abundant anion in mucus, at about 120mM, and provides the driving force for osmotic-induced changes in mucus volume. In contrast, HCO_3_^−^ concentration is about 30mM and may proportionally affect mucus volume. Additionally, HCO_3_^−^ may play an important role in buffering mucus pH. Mouse models with airway-specific overexpression of ENaC or exhibiting airway surface dehydration (mucus hyper concentration), impaired MCC and mucus plugging, mimic muco-obstructive lung disease sharing key features of CF and COPD [88].

Oxidative stress also influences COPD pathogenesis. Unlike lightly cross-linked mucus gels, the pathologic mucus in lung disease is not easily transported by the mucociliary escalator, causing its build up, lung infection atelectasis, and airflow obstruction [68,89]. In addition to the aforementioned cysteine-rich domains in mucin N and C termini, cysteine-rich regions are also abundant within internal domains. These internal cysteine thiols add to the antioxidant effects of mucins. High levels of oxidative stress in COPD mucus might increase mucin oxidative cross linkage through disulfide bonds, resulting in highly cross linked, elastic mucus [90]. Thus, antioxidant therapy may be effective against airway mucus hypersecretion.

### 4.3. Ciliophagy and CBF

Ciliophagy is a cilia-specific form of autophagy that mediates lung damage from cigarette smoking. It causes cilia shortening and mucus hypersecretion, hence airway epithelial damage [91]. Significant ciliary shortening has been seen in COPD patients with a history of chronic smoking, resulting in impaired mucociliary clearance. Previous studies have shown that lung tissue from COPD patients accumulates autophagy which negatively regulates the formation in cilia [92]. Lam and colleagues showed that elevated autophagy in primary cultured epithelial cells under cigarette smoke (CS) exposure was accompanied by cilia shortening and increased localization of ciliary proteins, including Ift88, Arl13, centrin 1, and pericentrin, to autophagosomes in mouse airways demonstrating that autophagy is increased in COPD cells under CS exposure, and that cilia shortening is caused by increased autophagic degradation of ciliogenic proteins. Additionally, genetic suppression of autophagy was found to be protective against CS-induced cilia shortening in vitro and in vivo [93].

## 5. Therapeutic Approaches and Targets of Mucus Hypersecretion

At present, drug therapy of airway mucus hypersecretion is mainly divided into 7 types, and the specific mechanism or targets and representative drugs are detailed in Table 2.

### 5.1. Mucus Hypersecretion Inhibitors Targeting Cell Differentiation

Various natural products exhibit potential therapeutic benefit against mucus hypersecretion. Most studies seeking to identify inhibitors of mucus hypersecretion target NF-κB [103]. Among natural mucoregulatory agents, S-Allylmercapto-L-cysteine (SAMC), Obtusifolin, Flavonoid 7,4′-Dihydroxyflavone(Flavonoid 7,4′-DHF), Luteolin and Platycodin D have been reported to be the most potent suppressors of MUC5AC expression in airway epithelial cells. A study by Min et al. found that SAMC, an organosulfur compound in garlic, modulates MUC5AC and aquaporin 5 (AQP5) expression in a COPD model by regulating the NF-κB signaling. SAMC treatment in the 20–100 μM range restored lipopolysaccharides (LPS) -induced suppression of IκBα expression levels in SPC-A1 cells in a dose-dependent manner [64]. A study by Byung-Soo Choi and colleagues using NCI-H292 cells found that 10–50 μM obtusifolin inhibits phorbol 12-myristate 13-acetate (PMA)-induced NF-κB nuclear translocation [94]. Moreover, Flavonoid 7,4′-DHF, a triterpenoid found in *G. uralensis,* and Luteolin, a flavonoidal found in *L. japonica* Thunb have exhibited similar effects on cell models, and significantly suppress MUC5AC expression and mucus production in PMA stimulated NCI-H292 airway epithelial cells [63,95].

The root of *P. grandiflorum*, a common medicinal and edible plant in East Asian, is widely used to calm panting in traditional Chinese medicine (TCM). Among its multiple active ingredients, including steroidal saponins, flavonoids, phenolic acids, and sterols, platycodin D(3) is thought to be main active compound against COPD. A Korean study by Seok JH et al. compared the effects of *P. grandiflorum* de Candolle (APG), platycodin D(3) and deapi-platycodin against airway mucin hypersecretion in PMA-stimulated NCI-H292 cells and sulfur dioxide-stimulated rat bronchitis pulmonary mucin hypersecretion in vivo. This study suggests that Platycodin D(3) could inhibit airway mucus hypersecretion through inhibition of MUC5AC production and promotion of MUC5AC secretion in NCI-H292 cells [96].

Acetylcholine, which is released by parasympathetic nerve fibers, is a classical neurotransmitter in airways that induces mucus secretion and bronchoconstriction via muscarinic receptors [104]. For this reason, anticholinergics are applied in the treatment of obstructive airway diseases [105]. The long-acting anticholinergic agent, tiotropium, has been proven to be effective against COPD. A role for acetylcholine in epithelial cell differentiation and goblet cell metaplasia has been proposed. Tiotropium inhibits ovalbumin-induced goblet cell metaplasia in mice and guinea pigs. Inhibition of endogenous acetylcholine by tiotropium does not affect epithelial cell differentiation after air exposure but inhibits and reverses IL-13-induced goblet cell metaplasia and MUC5AC expression. This effect of tiotropium is thought to be mediated via FOXA2 and FOXA3 [65]. A Chinese multicenter, randomized, double-blind, placebo-controlled clinical trial on patients with mild or moderate COPD treated with tiotropium, found that the annual forced expiratory volume in 1 s (FEV1) decline after bronchodilator use was significantly lower in the tiotropium-treated group relative to the placebo-treated group (29 ± 5 mL/year vs. 51 ± 6 mL/year) [97].

### 5.2. Mucus Hypersecretion Inhibitors Targeting Ion Transport

CFTR mutations cause cystic fibrosis (CF), an airway disease that phenotypically resembles COPD and is characterized by chronic bronchitis. Acquired CFTR dysfunction may influence COPD pathogenesis, making CFTR a potential anti-COPD target [81,82]. Ivacaftor, a CFTR activator, reverses CFTR dysfunction in vitro by activating wild-type CFTR-dependent, short circuit current following chronic exposure. A study by Raju et al. found that cigarette smoke extract (CSE) exposure reduces CFTR-dependent current in human bronchial epithelial (HBE) cells and human bronchi via a mechanism that may involve CSE-induced reduction in CFTR gating, decreasing CFTR open-channel. Ivacaftor, which reverses these adverse effects in vitro, is potentially beneficial for COPD patients with chronic bronchitis [67]. Roflumilast, a phosphodiesterase IV inhibitor, was the first oral COPD drug approved by the U.S. Food & Drug Administration (FDA). This drug was approved on 5 October 2018 for the treatment of COPD at all severities. Roflumilast significantly reduced the frequency of moderate and severe attacks in patients with previous hospitalization in large clinical trials [98]. An in vitro study by Schmid A et al. on differentiated, primary human bronchial epithelial cells found that roflumilast acts on CFTR to enhance the effects of salbutamol against reduced functional Cl^-^ conductivity caused by smoke. Moreover, roflumilast enhances forskolin-induced CBF stimulation in airway surface liquid (ASL) volume supplemented smoked [106].

Inhaled hypertonic saline (HS) is an effective therapy for muco-obstructive lung diseases. It aids epithelial lining fluid (ELF) hydration, which aids in mucocilliary clearance, improving MCC, FEV1, and quality of life [107]. A clinical study by Ramsey et al. showed that HS inhalation acutely reduces non-cystic fibrosis bronchiectasis mucus concentration by up to 25% [99]. Some anti-inflammatory and antioxidation mucolytics, including carbocisteine and N-acetylcysteine are widely used to treat respiratory diseases with phlegm production due to their ability to facilitate sputum elimination. A randomized placebo-controlled clinical trial by Zhong NS et al., involving 709 patients, found that relative to placebo, administration of 1500 mg carbocisteine daily for a year significantly reduced the number of exacerbations per (1.01 [SE 0.06] vs. 1.35 [SE 0.06]). Indicating that carbocisteine can reduce yearly exacerbation rates in COPD patients, which is more critical than mucolysis for long-term management of COPD [79]. A randomized, double-blind placebo-controlled trial by Zhong NS et al., involving 1006 patients found that in patients with moderate-to-severe COPD, long-term use of 600 mg N-acetylcysteine, twice daily, significantly reduces the frequency of AECOPD, especially moderately severe cases [100].

### 5.3. Mucus Hypersecretion Inhibitors Targeting Ciliary Wobble and Mucus Transport

Moderate-severe COPD is effectively managed using long-acting inhaled bronchodilators. The long-acting beta-agonists, salmeterol and formoterol, can clear mucus in patients or increase ciliary beat frequency in animal models [101]. An open-label, single-center, randomized, cross-over study by Thomas Meyer et al., involving 24 patients with mild-moderate COPD, found that a single daily dose of 12 mg formoterol enhanced mucus clearance within 14 days of treatment [102]. A trial by Piatti et al. evaluating the effects of salmeterol xynaphoate on CBF of nasal epithelium and rheological parameters of tracheobronchial mucus found that it induced ciliostimulation in COPD patients and significantly faster CBF [69].

## 6. Challenges Facing the Use of Mucus Hypersecretion Inhibitors in Humans and Future Directions in Drug Development

Accumulating evidence shows that mucus hypersecretion is a critical player in COPD, suggesting that drugs targeting mucus hypersecretion may be effective against COPD. Some common drugs, including tiotropium, roflumilast, HS, N-acetylcysteine, carbocisteine, salmeterol and formoterol, are currently used to treat mucus hypersecretion in COPD. However, mucus hypersecretion involves production, transport and secretion steps, that present multiple potential targets. Current strategies have limited capacity to inhibit mucus hypersecretion. For instance, several clinical trials have shown that carbocisteine and N-acetylcysteine reduce the frequency of AECOPD by approximately 28% and 33.7%, respectively [79,100]. However, side effects limit the widespread use of some drugs. For example, atomized inhalation of HS is associated with mild and transient risks including bronchospasm, cough, and breathlessness, limiting its use [108].

COPD causes high morbidity and mortality worldwide, highlighting the urgent need for effective therapeutic strategies. Thus, the search for novel drugs and targets remains an active area of research. Antibody therapies, such as human anti-IL-13 monoclonal antibody, targeting IL-13-STAT6-SPDEF signaling, may be effective against mucus hypersecretion [77].

Small molecules from natural products are an important resource of pharmaceutical ingredients. TCM or natural medicine is increasingly used and widely accepted for its advantages, including multiple active components, links and targets. Thus, it is crucial to explore TCM for mucus hypersecretion inhibitors.

## Figures and Tables

**Figure 1 molecules-25-04437-f001:**
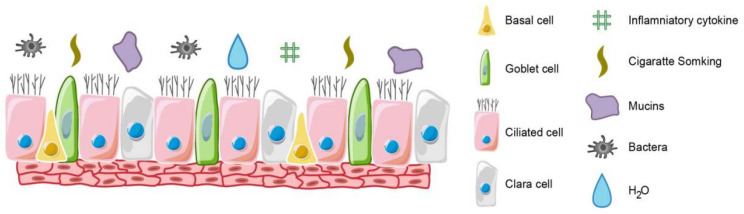
The structure of airway epithelium. It is a pseudostratified columnar epithelium containing basal, secretory and ciliated cells. Airway epithelial cell differentiation is influenced by factors such as bacterial infection, inflammatory factors and tobacco exposure.

**Figure 2 molecules-25-04437-f002:**
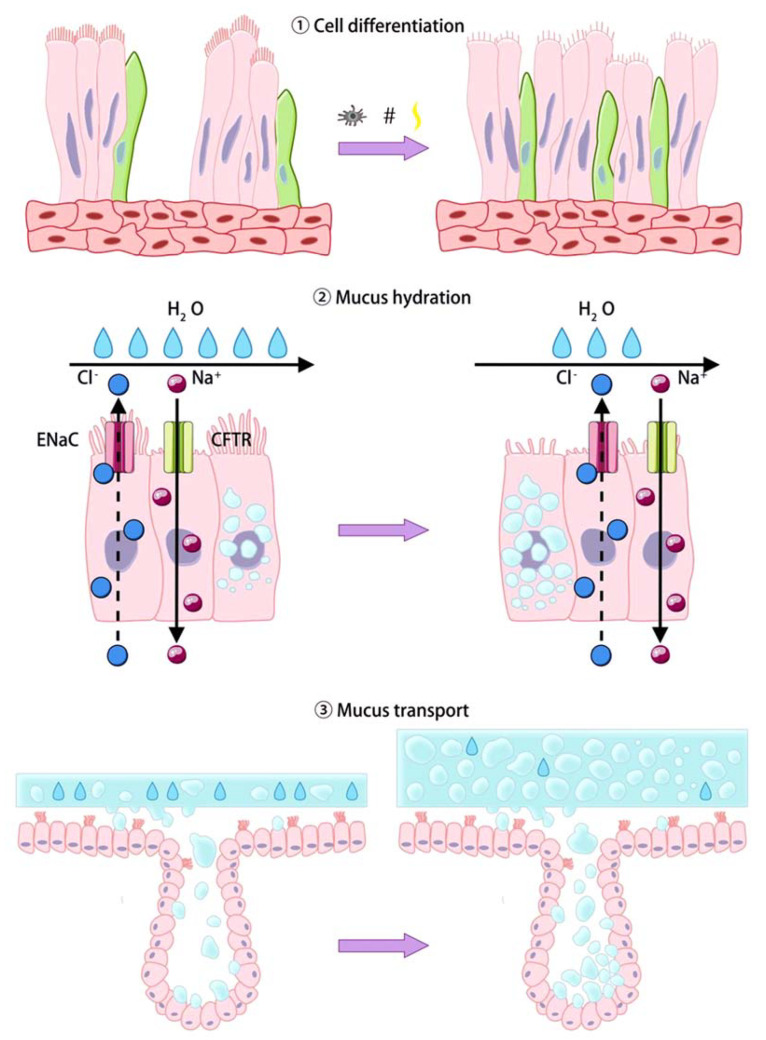
The process of mucin secretion and transport. There are three major steps in this process: (1) Goblet cell differentiation; (2) Dysfunction of chloride and sodium channels leads to mucus hydration; (3) Ciliated cell wobble and expel mucus.

**Figure 3 molecules-25-04437-f003:**
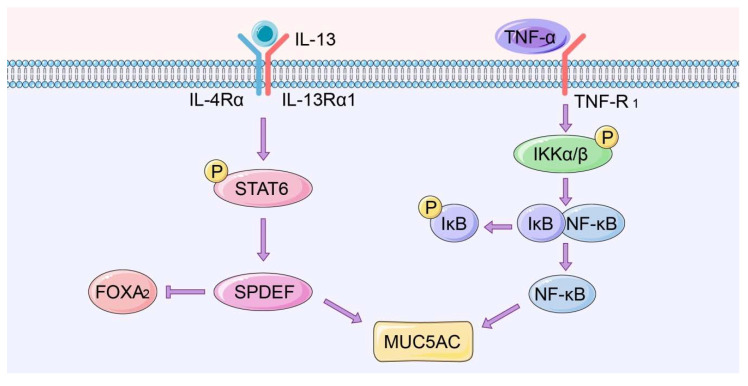
IL-13-STAT6-SPDEF and NF-κB signaling pathways associated with mucus hypersecretion.

**Table 1 molecules-25-04437-t001:** Mucin-type classification.

Character	Mucin	References
Membrane-constrained mucins (containing tandem repeats, tandem repeats TR)	Muc16, Muc15, Muc13, Muc12, Muc3B, Muc1, Muc3, Muc4, Muc17, Muc18 and Muc20	[29,30,31]
Cysteine-rich secretory mucins (containing TR sequences)	Muc19, Muc5B, Muc6, Muc5AC, and Muc2	[32,33]
Cysteine deficient secretory mucins (containing TR sequences)	Muc7, Muc8 and Muc9	[30]
Mucins without TR sequence mucins	Muc14, Muc15 and Muc18	[33,34,35,36]

**Table 2 molecules-25-04437-t002:** Representative drugs and targets for the treatment of mucus hypersecretion.

Link Acting on Airway Mucus Hypersecretion	Representative Compounds	TARGET or Mechanism	Reference
Epithelial Differentiation	S-Allylmercapto-L-cysteine *	NF-κB	[64]
	Obtusifolin *		[94]
	Flavonoid 7,4′-Dihydroxyflavone		[95]
	Luteolin *		[63]
	Platycodin D(3) *		[96]
	Tiotropium **	IL-13-STAT6-SPDEF	[65,97]
Ion Transport to Improve Hydration	Roflumilast **	CFTR	[98]
	Ivacaftor ***		[67]
	Hypertonic saline**	ENaC	[99]
	N-acetylcysteine **	Antioxidant	[100]
	Carbocisteine **	Thiol mucolytics	[79]
Mucus transport	Salmeterol and Formoterol **	Alter ciliary beat frequency	[69,101,102]

Note: (1) * Drugs in pre-clinical tests (2) ** most commonly used drugs. (3) *** Ivacaftor: limited to patients with CFTR mutations.

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
