# Peer review of "The Potential Role and Regulatory Mechanisms of MUC5AC in Chronic Obstructive Pulmonary Disease"

_molecules, 2020, doi:10.3390/molecules25194437_

Round 1
Reviewer 1 Report
Critical evaluation of the role of mucin 5AC in COPD is a worthy task and a well-organized comprehensive review will be a welcome addition to the COPD-related bibliography. This work may become such an addition, after a relatively straightforward improvements. The Authors need to very carefully revise and edit the work to weed out lapses that distract from the main topic and at some points make the text not understandable or misleading. Here are some examples.
The citations and references to citations in text need to be very carefully revised to weed our errors. For example, in the 4.1.2 chapter there is a reference to Hongmei Yu at al. [77], whereas position 77 is a work by Seibold. In turn, 2nd paragraph of the 4.1.1 chapter the start of the sentence should be: "A study by Soyound Kwak et al." not "A study Soyoung Kwa". Second paragraph of chapter 5.1 cites Jiho et al. - Jiho is the first name, not the second. The last two sentences of this paragraph likely were supposed to be presented as a single sentence - please fix.
First sentence in the third paragraph of Introduction is corrupted. Most likely, it should be "[...] tobacco smoking, exposure to biomass fuel, as well as genetic susceptibility."
Figure 2 is cited at the end of paragraph describing the Figure. It should be cited with the first mention of the Figure features.
Legend to Table 2: what "drugs in study phase" mean? Drugs in clinical trials? In pre-clinical tests? Both? Please specify.
Is TCM considered in COPD therapy beyond platycodin D3? If yes - please add information, if not yet - please speculate on opportunities beyond the statement that "it is crucial to explore".
Could the list of abbreviations be organized in alphabetical order? It is very difficult to use it now.
Please, revise the work - it can become a really useful review.
Reviewer 2 Report
This review is well structured and adds to the field, in the sense of summarising pathways relevant to this molecule and setting the context for drug development. Overall it was clear, and the figures were both professional and useful within the manuscript. Referencing was sufficiently comprehensive without being excessive. There are some minor changes which might improve the work.
- Page 2 'Mucins may be membrane bound or secreed' I think this is a typo of secreted
- Page 3 'MUC5AC... expression changes under stress' Could this be more specific? What sort of stress? Oxidative or something else?
- Figure 2 - it would be better to reference this within the text at the start of the sentences that it relates to, as I was confused by the listing of (left), (middle) etc in the text without any prior reference to the figure. eg 'Figure 2 demonstrates the processes of mucin secretion in health and disease
- Section 3.2 page 5 - it may not be true that all COPD patients have mucus hypersecretion etc, indeed most studies have shown that clinically apparent chronic bronchitis (ie expectoration of sputum) occurs in 40-60% of patients. Assuming it is present in asymptomatic people is not correct; suggest softening this statement.
- Section 3.3 page 5 Colonization is not the same as resistance; if you mean antibiotic resistance evidence to support this should be added
- Section 4.2 'Numerous studies have implicated physiological responses oxidation as a contributor to the damaging effects of oxidation' Is there a missing word in the sentence? Doesn't make sense
- Table 2 ivacaftor is specific to CF, so may not have a place in a COPD review. Please add a caveat to the table legend, or refer the reader to section 5.2 where this is discussed more
- Section 5.2 page 9, the trials of carbocisteine mentioned are only Chinese ones, and there have been both other trials and meta-analyses of this drug. Since this is a higher grade of evidence please also reference a systematic review
- Final section is numbered 5 but I think should read 6
Round 2
Reviewer 1 Report
Please, spell check! There are still typos well-visible even during fast reading. It distracts from the merits of the review.